
# Spatial indicators for desertification in south-east Vietnam

Le Thi Thu Hien[1], Anne Gobin[2], Pham Thi Thanh Huong[3]

[1] Institute of Geography, Vietnam Academy of Science and Technology (VAST), Hanoi, Vietnam
[2] Flemish Institute for Technological Research (VITO), Mol, Belgium
[3] The Vietnam Institute of Meteorology, Hydrology and Environment (IMHEN), Hanoi, Vietnam

*Correspondence to*: Anne Gobin (anne.gobin@vito.be)

**Abstract.** Desertification is influenced by different factors that relate to climate, soil, topography, geology, vegetation, human pressure and land and water management. The quantification of these factors into spatially explicit indicators and subsequent evaluation provides for a framework that allows to identify areas currently at risk of desertification and to

evaluate important contributing bio-physical and socio-economic factors. Applied to south-east Vietnam, a baseline 2010 map showed that 14.4% of the area, mainly along the coast and in the north east, is desertified with another 35.4% at severe risk of desertification. The Ministry of Environment has defined the area with a ratio of rainfall to evapotranspiration smaller or equal to 0.65, which equals 1,233 km$^2$ or 15% of the province, as desertified area. The developed framework allows for decision support in a what if structure, and for the projection of potentially vulnerable areas under future scenarios. With

projected climate change and current population growth the desertified area is expected to increase with 122% (or 137,850 ha) towards 2050. The methodology can be extended to neighbouring provinces that experience similar sensitivities to desertification.

## 1 Introduction

Desertification is "land degradation in arid, semiarid and dry sub-humid areas due to various factors, including climatic

variations and human activities" (UNCCD, 2012). Land degradation in this context means the progressive loss of land productivity (Geist, 2005). The United Nations Convention to Combat Desertification advocates methods to improve the global monitoring and assessment of dryland degradation to support decision-making in land and water management (UNCCD, 2012). Vietnam is highly affected by weather-related hazards that are projected to increase in frequency with climate change (Gobin et al., 2015). Vietnam is not designated as an arid or semi-arid country. However, the coastal

provinces in south-east Vietnam are strongly influenced by desertification, and therefore Vietnam ratified the convention in 1998 and formulated a national action programme in 2002.

The biophysical conditions related to desertification have originated models or frameworks involving the different processes of desertification to identify areas at risk (Schlesinger et al., 1990; Dirmeyer and Shukla, 1996; Kefi et al., 2007; Reynolds et al., 2007; Izzo et al., 2013; Jiang et al., 2019). Social, economic and in particular agricultural activities are considered as

important pressures impacting on land degradation and desertification (Okin et al., 2001; Asner et al., 2004; Zhou et al.,



2015; Hamidov et al., 2016). Integrating the monitoring and assessment of human and environmental variables poses major methodological challenges (Winslow et al., 2011) when assessing spatial desertification risks (Izzo et al., 2013; Zhou et al., 2015). Desertified areas have been detected using a multi-factorial approach such as the multi-component GIS framework for desertification risk assessment (Santini et al., 2010) or the Environmentally Sensitive Areas (ESA) approach (Kosmas et

al., 2006; Basso et al., 2000). Though originally designed for European Mediterranean environments, the ESA methodology has increasingly been used for classifying sensitivity to desertification in different environments ranging from Egypt (Gad and Lotfy, 2008), Iran (Parvari et al., 2011) and Central Asia (Jiang et al., 2019) to the Dominican Republic (Izzo et al., 2013) and northwest China (Zhou et al., 2015). All of these approaches combine regional modelling methods with spatially explicit information to assess the sensitivity of different areas to desertification.

Monitoring and assessment of the underlying drivers of land condition changes helps target remedial actions to alleviate true causes of land degradation (Gobin et al., 1999; Winslow et al., 2011; Zhou et al., 2015). The integration of socio-economic and bio-physical factors is included in several assessment frameworks such as the Dryland Development Paradigm (Reynolds et al., 2007; Stringer et al., 2017), the Monitoring and Assessment Indicator System (Zucca et al., 2012), Global Drylands Observing System (Verstraete et al., 2011; Bestelmeyer et al., 2015) and monitoring Sustainable Land

Management (Thomas, 2008; Hamidov et al., 2016). Focus on the integration of a particular component into a joint biophysical and socio-economic system is further elaborated for economic analysis (Salvati et al., 2008; Requier-Desjardins et al., 2011; Schild et al., 2018), institutional knowledge (Stringer et al., 2009) and mainstreaming policies on degradation (Akhtar-Schuster et al., 2011). Eliciting the underlying factors that cause drought and desertification is a prerequisite to the further establishment of a monitoring and assessment systems to support decision-making on land and water management.

The driving factors often relate to socio-economic developments with land use changes taken as valuable proxy indicators (Geist and Lambin, 2004; Hill et al., 2008; Hamidov et al., 2016). The observed relation between degradation and changes in ecosystem productivity have triggered the incorporation of remote sensing derived indicators (Cherlet et al., 2018; Jiang et al., 2019).

A multi-factorial approach was developed and based on local knowledge of environmental processes as documented in Hai

et al. (2013, 2014, 2016), thereby including socio-economic and bio-physical factors. We hypothesised that the area currently at severe risk of desertification can be explained by the different socio-economic and bio-physical factors. The objectives were to develop a framework that integrates biophysical and socio-economic factors contributing to the risk of desertification, spatially delineate areas at risk of desertification in the year 2010 as a baseline, and demonstrate the developed framework for potential vulnerable areas under projected climate change and population growth. The method was

developed for the Binh Thuan Province located in south-east Vietnam. The framework is designed to support policy makers in making informed decisions on combating desertification, analysing different scenarios and policy options, and formulating policies under what-if conditions.



## 2 South-east Vietnam

Vietnam has narrow deserts stretching along the central coastal areas, concentrated in 10 provinces from Quang Binh to Binh Thuan with a total area of about 419,000 ha. The U.S. Department of Agriculture recognised this problem in Vietnam in its world map of vulnerable desertification (USDA-NRCS, 2003). In August 1998 Vietnam ratified the UNCCD and formulated

priority areas for natural resources management. In 2006, the "Vietnamese Action Plan against desertification for the period 2006 – 2010 with an outlook to 2020" was adopted. Vietnamese studies on climate change and related environmental problems show that scientific, technological and policy responses are all needed for successful adaptation (Adger et al., 2005; Adger, 1999). Research has been implemented to study climate change phenomena and their effects such as sea level changes (Boateng, 2012; Zeidler, 1997), floods (Thi et al., 2010; Thanh et al., 2004), storms (Kleinen, 2007) and drought

(Sinha et al., 2011). Desertification, however, has received little attention in Vietnam.

Binh Thuan Province is located in the southern part of Central Vietnam, covers an area of 7,856 km² and has about 250 km of coastline. Binh Thuan and the neighbouring Ninh Thuan Province have a typical semi-arid climate with low rainfall, and high evaporation, with a variety of typical desert lands including sand, stone or salt deserts and degraded land. The Tuy Phong and Bac Binh districts (Figure 1) face 6 to 9 dry months per year with less than 100 mm of monthly rainfall. The

province is also subject to high temperatures and strong land winds that contribute to drought. The current desertification has a strong impact on overall production, environment and socio-economic activities.

As part of the National Action Plan to combat desertification, the Vietnam Ministry of Natural Resource and Environment (MONRE) delineated the desertified zone in Binh Thuan based on an aridity index, i.e. ratio of rainfall to evapotranspiration, smaller or equal to 0.65. The total desertified area is 1,233km², corresponding to 15% of the province (Figure 1).

Desertification, however, is the result of a variety of different factors and we therefore challenge the current delineation which is solely based on meteorological variables.

## 3 Data and Methods

Areas at risk of land degradation and desertification can be delineated using a multi-factorial approach based on the prevailing environmental processes that contribute to the risk. The processes are related to soil, climate, vegetation and land

management. Such a framework can be adapted to include soil, climate and vegetation as the main driving factors for desertification (e.g. Gad and Lotfy, 2008; Basso et al., 2012), incorporate local land use and vegetation characteristics (e.g. Izzo et al., 2013; Zhou et al., 2015; Jiang et al., 2019) or add projections of population densities (e.g. Salvati and Bajocco, 2011; Schild et al., 2018).

Based on the major factors that determine environmental sensitivity to desertification in the Binh Thuan Province of

Vietnam (Hai et al., 2013, 2014, 2016) we included indicators for climate, soil-landscape, vegetation, water resources management and human activities into a common framework. Local knowledge on these different contributing factors relate to resources management and pressures from human population densities. Each of these factors requires a different


methodology since different processes are involved that relate to land degradation and desertification. All indicators are subsequently compared to the current status of desertification and combined for a spatially and temporally explicit analysis of areas at risk of desertification. The proposed criteria for each indicator are implemented in a relational database, which allows for transparent definition, the incorporation of socio-economic information and the iterative refinement of the
selection rules in the future.

## 3.1 Database development

Spatial and/or temporal data were collected to characterise the climate, the soil-landscape, the vegetation, the water resources and the different human activities in the region. The resulting database contained information on soils (1:100,000); geology, hydro-geology, hydrology; topography; natural vegetation; land use / cover change; population; and, land and water
resources. Soil resources, their characteristics and FAO-WRB classification were obtained from the Department of natural resources and environment of the Binh Thuan Province. Land cover changes were derived from Landsat ETM in 1994/1995 and SPOT4 in 2009/2010. The water resources potential for household use and agriculture including irrigation was derived from the hydrogeological database that includes groundwater reserves and exploitation, surface water distribution and flow characteristics, and operational and planned irrigation systems. Human activities were evaluated from land use in 2005 and
2010, and the province's master planning for 2010-2020 including a vision on 2050 (GSO, 2012). Population density and settlement distribution were obtained from statistics at the community level. In- and outward migration to and from the province included tourist activities. Economic statistics on agricultural activities covered data on cultivated land, crop types, grassland and livestock. Meteorological data included daily rainfall and daily minimum, maximum and mean temperature for the period 1960-2010 obtained from the Vietnam Institute of Meteorology.

## 3.2 Calculation of indicators

The scoring and scaling is explained for five different quality indicators (*QI*): climate (*CQI*), soil (*SQI*), vegetation (*VQI*), water management (*WMQI*) and the different human activities and demographic pressures (*HQI*). Each Quality Indicator comprises different sub-indicators calculated for each km² grid cell of the province (Figure 2).

The Climate Quality Indicator (*CQI*) is based on the Aridity Index (*AI*) and calculated for 13 meteorological stations according to:

$$CQI_{1991-2010} = (P_{1991-2010} \cdot AI_{1991-2010} \cdot \Delta AI_{1991-2010})^{1/3}$$

$$\Delta AI_{1991-2010} = \frac{AI_{1981-2010} - AI_{1991-2010}}{AI_{1981-2010}}$$





$$AI_{year} = \frac{P_{year}}{PET_{year}}$$

Where $\Delta AI_{1991\text{-}2010}$ is the change of the aridity index relative to a longer period; $P$ is the yearly average annual rainfall and $PET$ is the annual average Modified Penman Monteith evapotranspiration, calculated for the periods 1991-2010 and 1981-

2010. Values for $AI$ below 0.35 indicate very severe desertification, between 0.35 and 0.75 desertification, and above 0.75 indicate no desertification (FAO-UNESCO, 1977). The change during 1991-2010 ($\Delta AI_{1991\text{-}2010}$) was quantified as the weighted departure from the average for the period 1981-2010. Values for $P$, $PET$, $AI$ and $\Delta AI$ were interpolated using a nearest neighbour interpolation in ArcGIS Spatial Analyst, and classified after a scoring system. The scoring for annual rainfall is 1 (low) for more than 2000 mm; 1.2 for 1500 - 2000 mm; 1.4 for 1000 - 1500 mm; 1.6 for 500 – 1000 mm; 1.8 for

250 – 500 mm; and 2 for annual rainfall below 250 mm. The aridity index ($AI$) has score 2 when below 0.35; score 1.6 for values between 0.35 and 0.75; score 1.3 for values between 0.75 and 1.20; and, score 1 for values above 1.20. The score is 1.1 for $\Delta AI$ values between -0.2 and 0; and 1 for values between 0 and 0.2.

The Soil Quality Indicator ($SQI$) comprises a weighting of soil fertility, salinity, slope, soil depth, soil type and the presence

of rocks. For each of the soil qualities a higher score indicates a higher vulnerability to desertification and degradation.

$$SQI_{2010} = \left(S_{texture} \cdot S_{depth} \cdot S_{slope} \cdot S_{rock} \cdot S_{salinity}\right)^{1/5}$$

The soil texture is derived from the World Reference Base for Soil Resources. Arenosols and leptosols are considered most

sensitive to desertification in the area and receive a score 2, followed by Salic Fluvisols (1.8), Thionic Fluvisols (1.6), Gleyi-Umbric Fluvisols (1.4), Acrisols (1.2), Ferralsols (1.1) and Luvisols (1). A shallow soil depth of less than 30 cm has a high score of 2, followed by 30 – 50 cm (1.5) and 50 - 100 cm (1.2); well-developed soils with a profile depth deeper than 100 cm receive score 1. A slope above 25° receives a score of 2; a score of 1.5 is given to slopes between 8° and 25°; slopes between 3° and 8° receive score 1.2; and, slopes below 3° score 1. The presence of salinity or rock fragments each receive score 1.2,

whereas their absence receives score 1.

The Vegetation Quality Indicator ($VQI$) comprises the normalised difference vegetation index ($NDVI$), changes in **NDVI** ($\Delta NDVI$) and vegetation type.

$$VQI_{2010} = (VT_{2010} \cdot NDVI_{2010} \cdot \Delta NDVI_{1995-2010})^{1/3}$$

The vegetation type ($VT$) was extracted from the 2010 forest classification database. The vegetation type is evaluated as having a high (2), medium (1.5) or low risk (1) to degradation. $NDVI_{2010}$ is the average km² value calculated from SPOT





Vegetation images in 2010. *NDVI* changes were calculated from Landsat TM on 1994/1995 and SPOT on 2009/2010 ($\Delta NDVI_{1995-2010}$). Both *NDVI* and vegetation types were scored according to their vulnerability to desertification and degradation. Dense forests and high *NDVI* values received lower scores than sparse forest cover or low *NDVI* values. An *NDVI*$_{2010}$ value below or equal to 0.1 receives a high scoring of 2; a value between 0.1 and 0.6 receives 1.5; and, a value

equal or above 0.6 receives a score of 1. $\Delta NDVI_{1995-2010}$ values above 0 are scored 2; values equal to 0 are scored 1.5; and, values below 0 are scored 1.

The Water Management Quality Indicator (*WMQI*) has three major components that include an assessment of the water use balance, groundwater capacity and irrigation characteristics.

$$WMQI_{2010} = (WB_{2010}.GW_{2010}.IT_{2010}.IC_{2010}.CD_{2010})^{1/5}$$

Where *WB* is the water use balance, *GW* is the groundwater capacity, *IT* is the irrigation type, *IC* is the irrigation capacity, and *CD* is the canal density. The water use balance (*WB*) is assessed from the volume of water extracted for irrigation and expressed as water short to meet irrigation demands in the perimeter and calculated from water demands for each agricultural

crop and water supply of each irrigation perimeter in 2010. A shortage in the water use balance of above $5.10^7\,\text{m}^3$ is scored 2, below $5.10^7\,\text{m}^3$ 1.5 and no shortage is scored 1. The assessment of the groundwater capacity (*GW*) is based on discharge data. Zero well discharges receive score 2, followed by discharges below 0.5 m³/h (1.6), 0.5-5.0 m³/h (1.3) and 5-10 m³/h (1). The irrigation characteristics are the irrigation type, the irrigation capacity in percentage of the area under irrigation and the length of irrigation canals per km² grid. Three irrigation types are distinguished: no irrigation with score 2, supplementary

irrigation with score 1.5 and full irrigation with score 1. For a zero canal density a score of 2 is assigned; a density between 0 and 0.25 km/km² gets a score of 1.5; and, a density above 0.25 receives 1. The irrigation capacity is scored as follows: 0 % (score 2); > 0-30% (score 1.6); 30-50% (score 1.4); 50% - <100% (score 1.2); and, 100% (score 1).

The human pressure and activities quality indicator is assessed on the basis of land use, land use change and population

density.

$$HQI_{2010} = (LU_{2010}.LUC_{2005-2010}.PD_{2010})^{1/3}$$

Where *LU* is the land use in 2010; *LUC* is the land use change between 2005 and 2010 and *PD* is the 2010 population

density. The classification and scoring focuses on vulnerability to desertification. The land use is scored: bare land (score 2); agriculture and settlements (score 1.5); and, forest and water (score 1). Land use changes to bare land are scored 2; changes of forest and water to other land uses are scored 1.5; and, the absence of changes are scored 1. A population density higher than 500 persons/km² gets score 2; densities between 200 and 500 receives score 1.5; and, densities below 200 have score 1.





All the above Quality Indicators are combined into one indicator for assessing the risk of each km² grid cell to desertification.

$$RI = (CQI.SQI.VQI.WMQI.HQI)^{1/5}$$

Where *RI* is the Risk Indicator, *CQI* is Climate Quality Indicator, *SQI* is Soil Quality Indicator, *VQI* is Vegetation Quality Indicator, *WMQI* is Water Management Quality Indicator and *HQI* is Human Pressure and activities Indicator.

We distinguished four major types of areas at risk based on a risk Indicator (*RI*). Critical areas include areas that are already

desertified and present a threat to the environment of the surrounding areas, with C3 being highly critical (*RI* > 1.53); C2 being medium critical ($1.42 \leq RI \leq 1.53$); and, C1 being low critical ($1.38 \leq RI < 1.42$). Fragile areas are areas in which any change in the balance of natural resources and human activities is likely to bring about degradation and desertification, with F3 highly fragile ($1.33 \leq RI < 1.38$); F2 medium fragile ($1.27 \leq RI < 1.33$); and, F1 low fragile: ($1.23 \leq RI < 1.27$). Potential area at risk are areas threatened by degradation under significant climate change, if a particular combination of land use is

implemented or where offsite impacts will produce severe problems elsewhere ($1.17 \leq RI < 1.23$). Areas not at risk are areas with a wet climate, well drained, well developed soils and/or with a dense vegetation cover; they are not considered as threatened by desertification (*RI* < 1.17).

### 3.3 Scenarios

A scenario of climate change and population growth was established as an example of how the framework can be used to support policy options. The high climate scenario (MONRE, 2009) was downscaled at the different meteorological stations using the MAGICC/SCENGEN software (Wigley, 2008). Human activities and population growth were taken from the master planning for 2010-2020 including a vision on 2050. The vision includes planned irrigation schemes which will in conjunction with climate change alter the water use balance. Based on governmental statistics (GSO, 2012) the province has

a population of 1.1 million people with a population growth of 1.4% during the last decade.

### 4 Results

### 4.1 Climate

The aridity index (*AI*) shows the occurrence of semi-arid regions in the northern coastal area (Figure 3). Based on *MI* comparison for different periods, drought is rising in the northern and the central coastal areas with up to 26% whereas in the

northwest of the province *MI* has increased with 10%. The value of *CQI* is high along the northern coastal area (Figure 3), stretching to the south and gradually moving to the east of the province indicating enhanced sensitivity to desertification.





With climate change, an increase of 10.4% in reference evapotranspiration is projected in the northwestern districts of Duc Linh, Tanh Linh and Ham Thuan Bac (Location see Figure 1; Table 1); a 12.2% increase in the southcentral districts of Ham Thuan Nam, Ham Tan and Phan Thiet; and, a 13.9% increase in the eastern districts of Tuy Phong and Bac Binh. The rainfall is projected to increase with up to 5% in the northwestern districts, remain the same in the central districts and

decrease with more than 10% in the northeastern districts. Overall this leads to a projected decrease in the ratio rainfall to evapotranspiration.

Climate is a key factor as confirmed by similar studies in the region (MONRE, 2009). The distinction between climate regimes and gradients within the province can be seen from the *MI* isolines (Figure 4). Other factors, however, play important roles and contribute to a much larger extent to mitigation and / or adaptation.

## 4.2 Soil and vegetation

The soil quality characteristics are evaluated with areas at risk displaying high *SQI* values (Figure 3). Poor soil quality due to shallow profile development, steep slopes or low fertility add to the risk. Deforested or eroded areas are mostly located in the mountains. Riverine areas with a good soil quality (low *SQI*, Figure 3) offer opportunities for farming or are vegetated with

dense forest. The soil quality indicator (*SQI*) provides for the clearest relation with desertification but cannot be interpreted on its own. Thin soils, steep slopes, vulnerability to erosion and sparse vegetation show a high risk for degradation in the mountains, whereas the plains and coastal areas are at risk due to salinisation and sandy soil textures.

Forests cover nearly half of the province and are located in the province's mountainous regions in the northwest and northeast. The vegetation quality indicator (*VQI*) displays a distinct pattern of low values (Figure 3) in the east where

mountains with dense-forests dominate the landscape and high values in the southeast and coastal areas where residential areas and agriculture are the principal land use. Low *NDVI* values indicate the presence of sparse vegetation with increased risks to degradation. At the same time afforestation or reforestation are important mitigation measure to combat desertification. *VQI* therefore reflects both adaptation and mitigation options.

## 4.3 Water Management

Binh Thuan Province has three major rivers with several tributaries mostly originating in the province itself or in the highlands of the neighbouring province. Two major lakes and several artificial lakes add to the surface water bodies that together with groundwater provide for water during the dry season. Water resources management such as the development of sustainable irrigation systems can alleviate drought and desertification. A water management quality indicator (*WMQI*) was

therefore included in the ESA methodology. The results show the effect of irrigation on the *WMQI* in the north east (Figure 3). The canal density and irrigation capacity determine the amount of fields that can be irrigated during the dry season. The *WMQI* reflects the availability of water resources and exploitation potential. Water supply and management have a large



impact on alleviating the negative effects of desertification (Figure 4), and therefore represent important adaptation measures.

## 4.4 Human pressure

The human pressure is evaluated using population density and land use intensity, the latter as a combination of land use and land use change (Figure 3). The population density ranges from around 1000/km² in Phan Thiet to less than 100/km² in the rural areas. As a result of strong population growth in the cities along the coast, urbanisation increased with 1% during the last decade, making Binh Thuan one of the most urbanised provinces of the South Central Coast in Vietnam.

The plains and coastal areas undergo a stronger human influence than the hills and mountains. Despite its large forested area,
the province has about 37% of agricultural land, which is the largest figure among all provinces of the central coast regions (GSO, 2012). One third of the agricultural land is used to cultivate rice. Though many mountains show clear signs of deforestation, there is an active policy towards reforestation and protection particularly of the dense forests in the northeast. The consequences of forest degradation may lead to further desertification and drought impacts on the coastal areas and plains. The human pressure and activities quality indicator (*HQI*) reflects the influence that population density can have on
natural resources use. Higher population pressures result in land use and vegetation changes, and give rise to a higher water demand for both agricultural and household use.

## 4.5 Areas at risk of desertification

The areas at risk of degradation in the Binh Thuan Province (Figure 3) shows a zoning of sensitivity with around 85% of the
province. Desertified areas account for 14.4% of the province, mainly the north-east and coastal areas. Another 35.4% is highly fragile and at immediate risk of desertification; this area is yearly affected by severe drought occurring during the dry season. In the northern districts the coastal zones suffer from shifting sand dunes and the plains from excessive drought and salinisation causing sparse vegetation and serious degradation. These districts experience a high risk of spreading desertification. The most vulnerable districts (Table 2; Figure 1) are located along the coastal areas of Tuy Phong (C: 370.9
km² or 49.3% of the district's area; F3: 288.1 km² or 38.3%), Ham Thuan Nam (C: 212.3 km² or 20.1%; F3: 503.1 km² or 47.7%), Bac Binh (C: 169.3 km² or 9.1%; F3: 686.7 km² or 36.9%) and Ham Thuan Bac (C: 127.2 km² or 49.3% of its area; F3: 486.6 km² or 38.3%) Districts. The densely populated coastal district of Phan Thiet (C: 102 km² or 51.8%; F3: 91.2 km² or 46.3%) is very vulnerable to moving sand dunes.

The trend of desertification under climate change and population growth is towards the southwest with a projected increase
of 1,379 km² land totalling 2,509 km² or 31.9% of the entire province being desertified towards 2050. The affected area is enlarging most in the districts of Ham Thuan Bac with 458 km², followed by Ham Tan with 258 km² and Ham Thuan Nam



with 210 km² (Table 2; Figure 1). The largest increase in desertification is expected in La Gi, Ham Thuan Bac and Ham Tan Districts where desertification is projected to increase with 490%, 360% and 277% respectively.

## 5 Discussion

According to MONRE the land affected by drought and desertification is about 43% of the Binh Thuan Province based on an aridity index below 0.8 which reflects the occurrence of a six-month dry season. A simple aridity index may provide a good indicator for the meteorological conditions but does not reflect the pressures nor the remedial actions that can be undertaken to alleviate the risk of desertification. In some areas irrigation systems to have been developed along with sustainable management practices such as adapted cropping systems and policies to protect water resources through reforestation schemes, e.g. in Bac Binh District.

The world vulnerability map of desertification (USDA-NRCS, 2003) presents the southern coastal zone of Vietnam as sensitive with the northern coastal zone of the Binh Thuan Province highlighted as highly sensitive. Despite the years of difference between our analysis and the USDA map the general picture is similar with the most sensitive areas being identified in both maps. In comparison to 1998 when Vietnam ratified the UNCCD, desertification has spread towards the southwest, a tendency that is continued under a scenario of climate change and population growth. Furthermore some areas in the centre have benefitted from irrigation schemes, whereas reforestation has taken place in parts of the northeast changing the sensitivity to desertification and justifying the incorporation of land and water management indicators in the approach.

Desertification and land degradation is the result of many factors, the characteristics of which differ for each region as for example in the Dominican Republic (Izzo et al., 2013), Italy (Basso et al., 2000), northwest China (Zhou et al., 2015), Central Asia (Jiang et al., 2019), Nigeria (Gobin et al., 1999) and Iran (Gad et Lotfy, 2008). In the Binh Thuan Province, different indices influence the environmental sensitivity in a different manner as demonstrated for five comprehensive quality indices, i.e. climate, soil, vegetation, water management and human pressure. Climate impacts will alter in the first place the climate quality indicator ($CQI$) mainly through temperature rises. Temperature rises affect evapotranspiration and the crop water demands, and the uncertain onset of the rainy season will affect crop water management and irrigation demands, both having an impact on water management ($WMQI$). The concept of the water footprint could help quantify crop water management (Gobin et al., 2017). Population growth alters the dynamics of land use demand and in turn affects water demands and water use balances. Land and water management seems a priority policy domain for alleviating the problems related to land degradation and desertification in the area.

The combination of different indices into the Risk Indicator ($RI$) allows for identifying zones at risk of desertification and for eliciting the most important factors influencing desertification; and provides for a method that is applicable across different regions in the world (e.g. Izzo et al., 2013; Jiang et al., 2019). Monitoring the different spatial indicators and quality indices is an important part of an approach to combat desertification and allows for informed decision making (Winslow et al., 2011). Subsequent studies could enlarge the area to regions at current or potential risk, and could benefit from other land





degradation related risk assessment research (Gobin et al., 1999; Kefi et al., 2007). The spatially explicit regional indices provide a comparative overview (Reynolds et al., 2007) and define areas where more detailed studies or policy attention are needed for determining the contributing factors and mitigating risks (Salvati et al., 2008; Hamidov et al., 2016). With changing scales, the dominant controlling factors such as for example land management, water resources development and

vegetation cover may change (Gobin et al., 2001; Santini et al., 2010; Zhou et al., 2015). Individual farming practices may be different too such that remediation strategies at the community level must be adapted to this scale. Monitoring at multiple scales therefore provides for a nested strategy of focussing on environmentally sensitive areas (Gobin et al., 2004) which may require remedial measures to be taken at different levels of decision making.

## 6 Conclusions

The approach of identifying environmental sensitive areas allows for describing current desertification and projecting trends. The area at risk of desertification in the Binh Thuan Province of south-east Vietnam is expected to increase from 14.4% in 2010 to 31.9% in 2050 under projected climate change and population growth.

The combination of several indicators and multi-criteria analysis proved useful for eliciting relationships between factors and identifying environmentally sensitive areas now and under environmental change. The key factors of desertification are

drought, soil type, human impact on land cover and irrigation water capacity. Many of these factors can be influenced through policies encouraging the sustainable use and management of natural resources. The method can be extended to neighbouring areas experiencing similar environmental threats.

### Acknowledgements

The authors would like to thank the Vietnam Academy of Science and Technology (VAST.DLT.13/13-14) and the Belgian

Science Policy Office (BELSPO BL/03/V28) for funding the research under an umbrella bilateral cooperation agreement between Vietnam and Belgium. The authors are indebted to the People's Committee of the Binh Thuan Province; the Department of Science and Technology; the Department of Natural Resources and Environment; the Institute of Meteorology, Hydrology and Environment; and, the Institute of Geography for lending data and scientific reports.

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

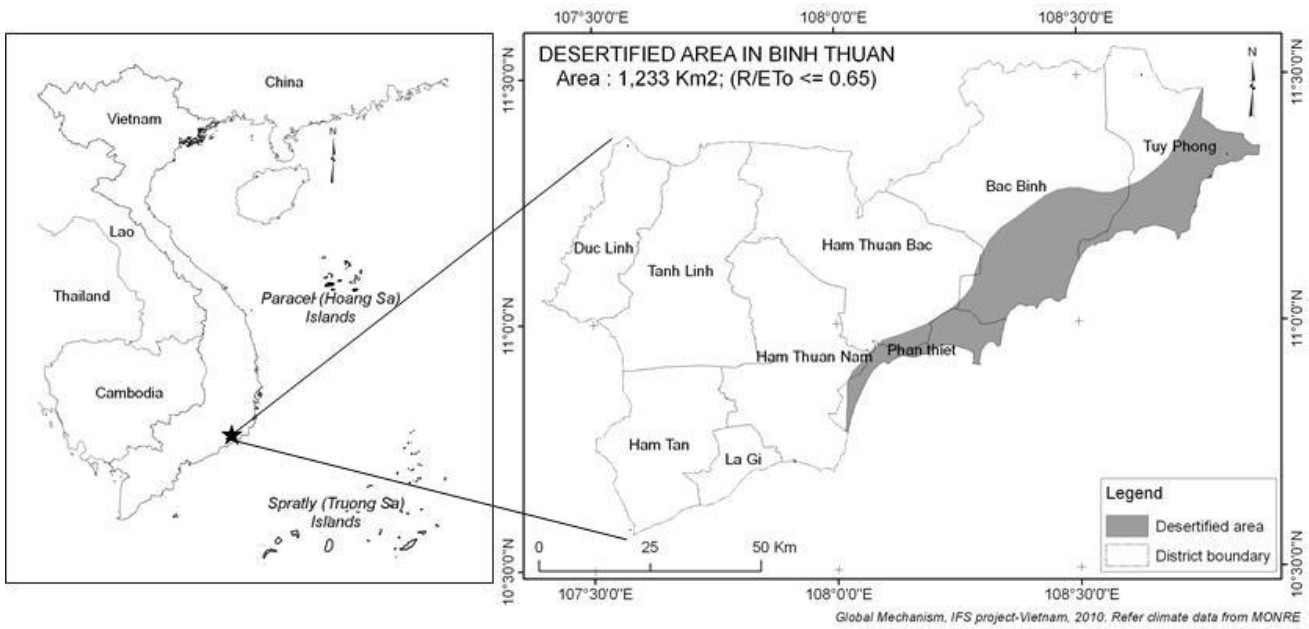

**Figure 1: Location of the desertified area in Binh Thuan Province in Vietnam, based on rainfall to evapotranspiration smaller or equal to 0.65.**



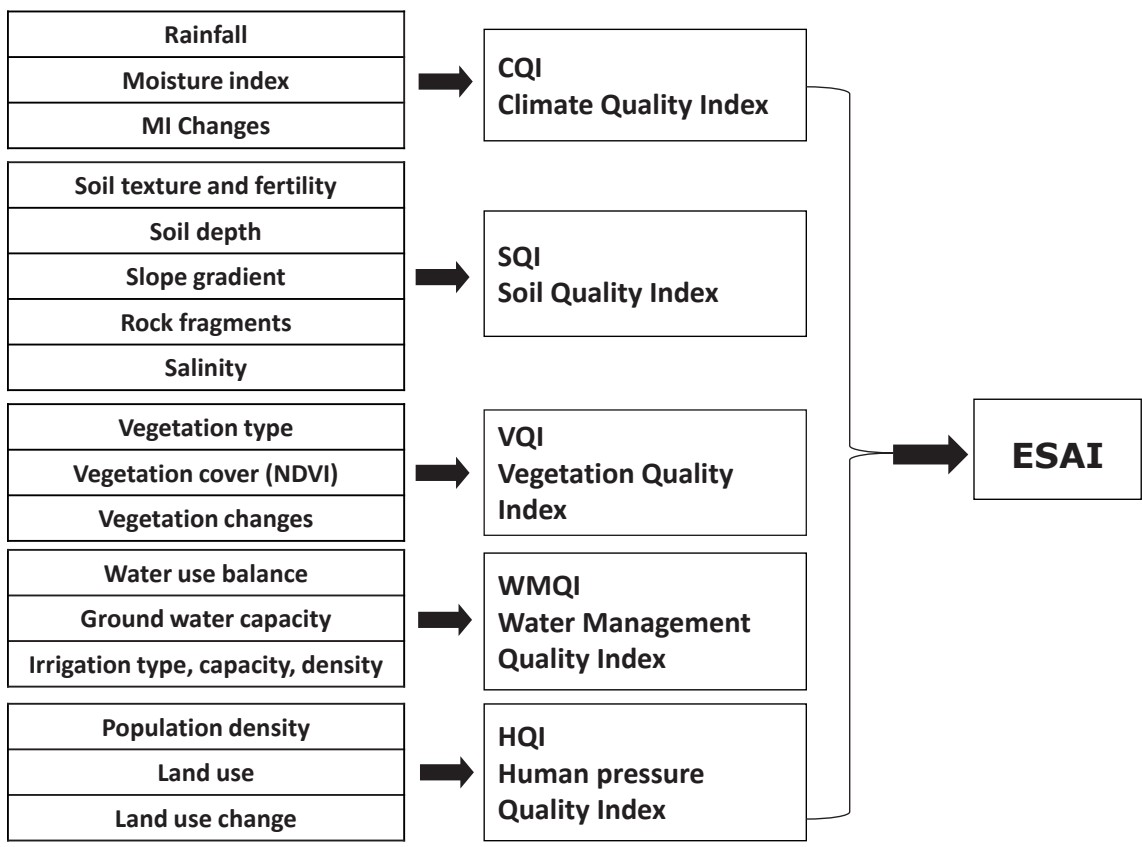

**Figure 2: Spatially distributed indices for identifying environmentally sensitive areas to desertification.**


**Figure 3: Results of the Climate Quality Indicator (CQI), Soil Quality Indicator (SQI), Vegetation Quality Indicator (VQI), Water Management Quality Indicator (WMQI), Human Pressure Indicator (HQI) and the Environmentally Sensitive Area Indicator (ESAI).**

**Figure 4: Major influencing quality indices on the Environmentally Sensitive Areas (ESA) in 2010 (top) and projections of ESA for 2050 (bottom) under climate change and population growth.**





**Table 1: Rainfall to reference evapotranspiration for 13 meteorological stations in the Binh Thuan Province for observed 1981-1990 and 1991-2010 periods and projected for 2050 according to MONRE's high climate scenario for Vietnam.**

| District | Station | LON | LAT | 1981-1990 (P/ET0) | 1991-2010 (P/ET0) | 2050 (P/ET0) |
|---|---|---|---|---|---|---|
| Tuy Phong | Lien Huong | 108°43' | 11°13' | 0.23 | 0.35 | 0.27 |
| Tuy Phong | Song Mao | 108°30' | 11°15' | 0.33 | 0.48 | 0.37 |
| Bac Binh | Bau Trang | 108°25' | 11°04' | 0.30 | 0.35 | 0.29 |
| Phan Thiet | Mui Ne | 108°17' | 10°56' | 0.40 | 0.44 | 0.38 |
| Phan Thiet | Phan Thiet | 108°06' | 10°55' | 0.50 | 0.55 | 0.49 |
| Ham Thuan Bac | Dong Giang | 108°00' | 11°13' | | 0.97 | 0.91 |
| Ham Thuan Nam | Ma Lam | 108°03' | 11°06' | 0.39 | 0.53 | 0.45 |
| Ham Tan | Ham Tan | 107⁰49' | 10⁰41' | 0.78 | 0.74 | 0.75 |
| Tanh Linh | La Ngau | 107°47' | 11°10' | 0.98 | 1.07 | 0.95 |
| Tanh Linh | Ta Pao | 107°46' | 11°07' | 1.19 | 1.09 | 1.03 |
| Tanh Linh | Suoi Kiet | 107°42' | 11°07' | | 0.93 | 0.87 |
| Duc Linh | Me Pu | 107°37' | 11°13' | 1.31 | 1.27 | 1.15 |
| Duc Linh | Vo Xu | 107°36' | 11°11' | 1.17 | 1.09 | 1.02 |

Where P is rainfall and ET0 is the reference evapotranspiration calculated according to the modified Penman Monteith method; For location of the districts see Figure 1.




**Table 2: Surface area covered by areas at risk to desertification in the Binh Thuan Province in 2010 and projections for 2050.**

| District | ESA type (km²) in 2010 | | | ESA type (km²) in 2050 | | |
|---|---|---|---|---|---|---|
| | N + P | F | C | N + P | F | C |
| Phan thiet | 0.0 | 95.1 | 102.0 | 0.0 | 8.3 | 188.8 |
| Tuy Phong | 0.0 | 381.1 | 370.9 | 0.0 | 271.4 | 480.6 |
| Bac Binh | 6.8 | 1684.4 | 169.3 | 34.4 | 1541.9 | 284.2 |
| Ham Thuan Bac | 37.0 | 1199.3 | 127.2 | 82.7 | 695.7 | 585.1 |
| Ham Thuan Nam | 13.3 | 828.5 | 212.3 | 0.9 | 630.9 | 422.3 |
| La Gi | 0.0 | 156.5 | 21.1 | 0.0 | 53.2 | 124.4 |
| Ham Tan | 0.0 | 635.2 | 93.2 | 0.0 | 376.9 | 351.5 |
| Duc Linh | 33.9 | 497.0 | 3.2 | 13.4 | 516.0 | 4.7 |
| Tanh Linh | 159.2 | 998.5 | 31.3 | 113.9 | 1007.8 | 67.3 |
| Total | 250.3 | 6475.5 | 1130.5 | 245.3 | 5102.0 | 2509.0 |
| % of total area | 3.2 | 82.4 | 14.4 | 3.1 | 64.9 | 31.9 |

Where N= not affected; P= potential; F= Fragile; C= Critical. Location see Figure 1.

