# Peer review of "Spatial indicators for desertification in south-east Vietnam"

_Natural Hazards and Earth System Sciences, 2019_

## Referee Comment (RC1) · Anonymous Referee #1 · 17 Jun 2019

The manuscript describes the application of a set of combined spatial indicators (multi-factorial approach) to describe desertification status and future trends in selected vunerable regions of Vietnam. The paper is in general well structured and written.

Few minor revisions as follows are suggested: -page 5 top: provide a reference for the applied Penman Montehith approach -page 5 top: i doubt that the climatic AI indicator alone does indicate desertification rather than the "risk of desertification", as used also for the other single indicators. Please modify it.

- page 5 ff: The scoring system of indicators remains unclear: explain for all individual indicators on which basis the scoring was defined. Is it artifical/subjective classification or was it calibrated? if yes, how it was done? is it base on other studies (provide references)?

[Figure]

-page 7: provide more information of applied climate scenarios: e.g. RCP type, mean temperature change, time slice....

-page 8 top: which type of reference evapotranspiration? (provide reference, e.g. is it FAO grass reference?)

-Figure 2 and 4: is the ESAI and ESA the same as the RI (see page 7 top) ? This is unclear. Please explain the difference or harmonize the terms, otherwise it remains confusing.

-Discussion: Please add a short description of uncertainties and limitations of the study and reserachs needs /gaps.

---

## Short Comment (SC1) · 17 Jul 2019

This is a well documented paper. It presents a methodology for assessing desertification in south-east Vietnam. This methodology has been successfully presented in the Mediterranean region previously. The paper constitutes and thorough and detailed approach for desertification identification and assessment. There are two questions-clarifications to be provided by the authors: 1. A clear distinction should be made in the text between RI and ESA to avoid any confusion. 2. An explanation is necessary for multi-criteria analysis, if it is applied, how and at what stage in the paper.

---

## Referee Comment (RC2) · Anonymous Referee #2 · 23 Jul 2019

I. General comments The manuscript constitutes an application procedure to describe desertification status and future trends in south Vietnam by using a set of combined spatial indicators (multifactorial approach). This methodology is not new and has been successfully applied in the Mediterrranean. The paper consists of a thorough and well documented application and is in general well structured and written.

II. Specific comments: A few minor revisions to be considered: 1. An explanation and clarification is necessary for ESAI and ESA: are the same? Also are the same with RI? If not, then specify the differences. This reflects Figures 2 and 4. 2. An explanation is necessary for muli-criteria analysis and how it is connected with the whole methodology. 3. There is a need to clarify the scoring system of individual indicators. Is it objective or subjective? 4. It would be useful if the authors could add

any limitations and uncertainties of the methodology.

---

## Referee Comment (RC3) · Anonymous Referee #3 · 30 Jul 2019

The manuscript presents a desertification risk indicator based on a simple scheme applied to a coastal region in the Vietnamese southeast. The results of the scheme are very similar to those of the World vulnerability map of desertification, page 10, lines 10-13, what question the interest of the manuscript.

The defined quality indicators are difficult to understand: 1. The climate quality indicator (CQI) is based on the average annual precipitation and reference aridity index and its temporal rate of change, neglecting the influence of other meteorological factors such as the wind, (the occurrence of strong land winds is mentioned in page 3, line 15), and the relative humidity. Why do not integrate these effects in a water balance in the air above the ground? Is there any rationale behind the selected threshold values used for the scores of the precipitation and aridity?

[Figure]

Incidentally, the scores for the different magnitudes should have been better indicated in a table.

2.-The soil quality indicator (SQI) includes the slope which is not properly an edaphic attribute. The texture scores should be based on the textural components, not on the units of a soil classification system what implies the contribution of other edaphic factors like rock presence, salinity, or depth, considered in other parts of the SQI. As in the previous indicator, the authors should have justified the limits between different categories. Why the presence of rocks and salinity are not better delimited?

3.-The vegetation quality indicator (VQI) is loosely defined. Is the vegetation of the study region so homogeneous that does not require any specification of trees, shrubs, or herbaceous plants? Is it necessary to include both the NDVI and is time rate of change at the same level in the VQI?

4.-The water management quality indicator (WMQI) is a mixture of very heterogeneous factors with the same level of influence. The water balance is not the volume of water used for irrigation. This volume should be expressed as volume per unit area to extend its potential use out of the study area. The groundwater capacity refers more precisely to a volume than to a discharge rate. The irrigation factors type and capacity are not similar as they appear in the WMQI equation. What relevance the canal density in the indicator? The existence of canals do not necessarily imply that they are in use.

The risk indicator demands a sound justification.

There are some formal aspects in addition to the convenience of tables to show the different scores for indicators and their factors:

1.-Is there a necessity to reinforce some of the statements with a host of references? The abundance of multiple references might be more an obstacle than a help for the reader.

2.-Some sentences are rather obvious (e.g. page 2 lines 25-26; page 3 lines 24-25;

page 3 lines 31-32, page 4 lines 1-3; page 11 lines 14-15).

3.-Some references are missing in the final list as the FAO-UNESCO of page 5 line 6-

---

## Author Comment (AC1) · 7 Sep 2019

Reply to Reviewer 1: We thank Reviewer 1 for his comments and suggestions. Below we document how we have changed our manuscript accordingly.

- page 5 top: provide a reference for the applied Penman Montheith approach A reference to Allen et al. (1998) has been added.

- page 5 top: I doubt that the climatic AI indicator alone does indicate desertification rather than the "risk of desertification", as used also for the other single indicators. Please modify it. Absolutely correct and we have changed this accordingly.

- page 5 ff: The scoring system of indicators remains unclear: explain for all individual indicators on which basis the scoring was defined. Is it artificial/subjective classification
or was it calibrated? if yes, how it was done? is it based on other studies (provide references)? The scoring of the quality indicators was based on a multi-factorial approach combining the multi-component GIS framework for desertification risk assessment by Santini et al. (2010) and the Environmentally Sensitive Areas (ESA) approach by Kosmas et al. (2006) and Basso et al. (2000).

- page 7: provide more information of applied climate scenarios: e.g. RCP type, mean temperature change, time slice.... We added a paragraph on the climate scenario that we used: "During the period 1958-2007, the average temperature increased by 0.5–0.7°C. Vietnam's official scenarios for climate change (MONRE, 2009) fit these current trends. The medium emission scenarios corresponds to an increase in temperature of 1°C by 2050 and 2.4°C by 2100 with respect to the 1980-1999 baseline period. Rainfall in the middle of the rainy season would increase 10-15% with respect to the 1980-1999 baseline period in the South Central. On a year basis, rainfall is projected to increase with 1.7% by 2050 and 3.2% by 2100. This climate scenario corresponds to RCP4.5 with a radiative forcing of 4.5 W/m$^2$ and 650 ppm CO2 equivalent in 2100."

- page 8 top: which type of reference evapotranspiration? (provide reference, e.g. is it FAO grass reference?) The Penman-Monteith evapotranspiration (Allen et al., 1998) was used. This has now been documented and referenced to in the manuscript.

- Figure 2 and 4: is the ESAI and ESA the same as the RI (see page 7 top) ? This is unclear. Please explain the difference or harmonize the terms, otherwise it remains confusing. We agree. We have harmonized the manuscript and used ESAI throughout the document to denote the Environmentally Sensitive Area Indicator.

- Discussion: Please add a short description of uncertainties and limitations of the study and research needs /gaps. We added a short description in the discussion section, marked in yellow.

Please also note the supplement to this comment:

https://www.nat-hazards-earth-syst-sci-discuss.net/nhess-2019-146/nhess-2019-146-AC1-supplement.pdf

---

## Author Comment (AC2) · 7 Sep 2019

Reply to Reviewer N. Dalezios Thank you for the review, Prof. Dalezios.

There are two questions clarifications to be provided by the authors:

1. A clear distinction should be made in the text between RI and ESA to avoid any confusion. This comment is very pertinent and the confusion has been removed from the manuscript. We have opted for the very well-known terminology ESAI (Environmentally Sensitive Area Indicator).

2. An explanation is necessary for multi-criteria analysis, if it is applied, how and at what stage in the paper. We have used a factorial approach and gave each factor equal weight in the Quality Indicators (QI), and in turn in the Environmentally Sensitive
Area Indicator (ESAI). We did not use multi-criteria analysis in this research, but the suggestion is very valid and we have added the idea to the discussion section.

---

## Author Comment (AC3) · 7 Sep 2019

Reply to Reviewer #2

We thank reviewer 2 for the positive general comments.

A few minor revisions to be considered: 1. An explanation and clarification is necessary for ESAI and ESA: are the same? Also are the same with RI? If not, then specify the differences. This reflects Figures 2 and 4. The confusion has been removed from the manuscript. We have opted for the known terminology ESAI (Environmentally Sensitive Area Indicator), and have adapted the text and figures accordingly.

2. An explanation is necessary for muli-criteria analysis and how it is connected with the whole methodology. We have used a factorial approach and gave each factor

equal weight in the Quality Indicators (QI), and in turn in the Environmentally Sensitive Area Indicator (ESAI). We did not use multi-criteria analysis in this research, but the suggestion is very valid and we have added the idea to the discussion section.

3. There is a need to clarify the scoring system of individual indicators. Is it objective or subjective? The scoring of the quality indicators was based on a multi-factorial approach combining the multi-component GIS framework for desertification risk assessment by Santini et al. (2010) and the Environmentally Sensitive Areas (ESA) approach by Kosmas et al. (2006) and Basso et al. (2000). The scoring system as presented here, and adopted by a lot of studies is subjective and could benefit further from data mining techniques.

4. It would be useful if the authors could add any limitations and uncertainties of the methodology. We have incorporated a section on the limitations of the methodology in the discussion section.

Please also note the supplement to this comment:
https://www.nat-hazards-earth-syst-sci-discuss.net/nhess-2019-146/nhess-2019-146-AC3-supplement.pdf

---

## Author Comment (AC4) · 7 Sep 2019

Reply to Reviewer #3

We thank Reviewer 3 for a thorough review and for highlighting interesting discussion points on the methodology used. Many of the comments raised in this review have helped reshape the discussion section or enabled a better explanation or justification of the methodology section.

The defined quality indicators are difficult to understand: 1. The climate quality indicator (CQI) is based on the average annual precipitation and reference aridity index and its temporal rate of change, neglecting the influence of other meteorological factors such as the wind, (the occurrence of strong land winds is mentioned in page 3, line

15), and the relative humidity. Why do not integrate these effects in a water balance in the air above the ground? Is there any rationale behind the selected threshold values used for the scores of the precipitation and aridity? Incidentally, the scores for the different magnitudes should have been better indicated in a table. We have made the text clearer and clarified our methodology better. As the reviewer rightly points out there was confusion in the initially submitted manuscript. The initial formula of the aridity index includes temperature based evapotranspiration according to Thornthwaite (1948). When using the modified Penman-Monteith equation (Allen at al., 1998), wind and humidity are incorporated. Both wind and relative humidity are important contributors to evapotranspiration, which is together with rainfall taken into account in the aridity index. An important improvement could indeed be a water balance and the incorporation of other variables in the climate quality indicator. We have taken up these points in the discussion section.

2. The soil quality indicator (SQI) includes the slope which is not properly an edaphic attribute. The texture scores should be based on the textural components, not on the units of a soil classification system what implies the contribution of other edaphic factors like rock presence, salinity, or depth, considered in other parts of the SQI. As in the previous indicator, the authors should have justified the limits between different categories. Why the presence of rocks and salinity are not better delimited? We concentrated on pedological properties of soil development, and hence the choice for soil classification related properties: the presence of rocks, salinity, profile depth, soil texture and slope. The inclusion of edaphic properties is a very valid comment, which we have taken up in the discussion. However, this suggested approach requires soil data at more sampling points than currently available in the study region.

3. The vegetation quality indicator (VQI) is loosely defined. Is the vegetation of the study region so homogeneous that does not require any specification of trees, shrubs, or herbaceous plants? Is it necessary to include both the NDVI and is time rate of change at the same level in the VQI? Correct and we provided further clarification. The

forest classification geo-database also includes other natural vegetation classes, ranging from broad-leaved evergreen humid forest to secondary natural dune vegetation. NDVI values are an indication of vegetation greenness and health; a declining change in NDVI indicates degradation.

4. The water management quality indicator (WMQI) is a mixture of very heterogeneous factors with the same level of influence. The water balance is not the volume of water used for irrigation. This volume should be expressed as volume per unit area to extend its potential use out of the study area. The groundwater capacity refers more precisely to a volume than to a discharge rate. The irrigation factors type and capacity are not similar as they appear in the WMQI equation. What relevance the canal density in the indicator? The existence of canals do not necessarily imply that they are in use. We clarified the explanation of the water use balance calculation. The water use balance is expressed per irrigation perimeter, and reflects the balance between demand and supply. Irrigation water supply discharges were provided by the water board, and also cropping areas but no exact location of the crops; hence the choice to categorize and score the different perimeters. The canal density refers to used canals, which were checked during field surveys in 2010. We have checked the explanations of the WMQI to clarify the calculations that we performed.

5. The risk indicator demands a sound justification. There are some formal aspects in addition to the convenience of tables to show the different scores for indicators and their factors: a. Is there a necessity to reinforce some of the statements with a host of references? The abundance of multiple references might be more an obstacle than a help for the reader. We agree. To avoid this confusion we have deleted the first paragraph of data and methods. The references and statements have been sufficiently covered in the introduction.

b. Some sentences are rather obvious (e.g. page 2 lines 25-26; page 3 lines 24-25; page 3 lines 31-32, page 4 lines 1-3; page 11 lines 14-15). We agree! Page 2 lines 25-26; Page 3 lines 24-25; Page 3 lines 31-32: a sentence has been removed. Page 4

lines 1-3; page 11 lines 14-15 have been rephrased to remove the rather obvious.

c. Some references are missing in the final list as the FAO-UNESCO of page 5 line 6- We have cross-checked the references; and added the missing reference to FAO-UNESCO-WMO.

Please also note the supplement to this comment:
https://www.nat-hazards-earth-syst-sci-discuss.net/nhess-2019-146/nhess-2019-146-AC4-supplement.pdf

[Figure]

**Supplement:**

[revised manuscript text omitted]